# Blood n-3 fatty acid levels and total and cause-specific mortality from 17 prospective studies

William S. Harris[1,2 ✉], Nathan L. Tintle [2,3], Fumiaki Imamura [4], Frank Qian[5,6], Andres V. Ardisson Korat[6], Matti Marklund[7,8], Luc Djoussé[6], Julie K. Bassett[9], Pierre-Hugues Carmichael[10], Yun-Yu Chen[11], Yoichiro Hirakawa[12], Leanne K. Küpers[13], Federica Laguzzi[14], Maria Lankinen[15], Rachel A. Murphy[16], Cécilia Samieri[17], Mackenzie K. Senn[18], Peilin Shi[19], Jyrki K. Virtanen [15], Ingeborg A. Brouwer[20], Kuo-Liong Chien[21,22], Gudny Eiriksdottir[23], Nita G. Forouhi [4], Johanna M. Geleijnse [13], Graham G. Giles [24], Vilmundur Gudnason [23,25], Catherine Helmer[17], Allison Hodge [24], Rebecca Jackson[26], Kay-Tee Khaw[4], Markku Laakso [27], Heidi Lai[19,28], Danielle Laurin[10,29], Karin Leander[14], Joan Lindsay[30], Renata Micha[19], Jaako Mursu[15], Toshiharu Ninomiya[31], Wendy Post[9], Bruce M. Psaty[32], Ulf Risérus[33], Jennifer G. Robinson[34,35], Aladdin H. Shadyab[36], Linda Snetselaar[35], Aleix Sala-Vila[2,37], Yangbo Sun[35,38], Lyn M. Steffen[39], Michael Y. Tsai[40], Nicholas J. Wareham [4], Alexis C. Wood[18], Jason H. Y. Wu[7], Frank Hu[5,6], Qi Sun [5,6], David S. Siscovick[41], Rozenn N. Lemaitre [32], Dariush Mozaffarian [19] & The Fatty Acids and Outcomes Research Consortium (FORCE)*

The health effects of omega-3 fatty acids have been controversial. Here we report the results of a de novo pooled analysis conducted with data from 17 prospective cohort studies examining the associations between blood omega-3 fatty acid levels and risk for all-cause mortality. Over a median of 16 years of follow-up, 15,720 deaths occurred among 42,466 individuals. We found that, after multivariable adjustment for relevant risk factors, risk for death from all causes was significantly lower (by 15–18%, at least $p < 0.003$) in the highest vs the lowest quintile for circulating long chain (20–22 carbon) omega-3 fatty acids (eicosa-pentaenoic, docosapentaenoic, and docosahexaenoic acids). Similar relationships were seen for death from cardiovascular disease, cancer and other causes. No associations were seen with the 18-carbon omega-3, alpha-linolenic acid. These findings suggest that higher circulating levels of marine n-3 PUFA are associated with a lower risk of premature death.

A full list of author affiliations appears at the end of the paper.

The n-3 polyunsaturated fatty acid (PUFA) family has been the subject of intense investigation ever since their inverse associations with risk for acute myocardial infarction were reported in Greenland Eskimos in the 1970s[1,2]. The PUFAs in this family include the 18-carbon, plant-derived alpha-linolenic acid (ALA,) as well as the 20–22-carbon, long-chain (LC, mostly seafood-derived) eicosapentaenoic (EPA), docosapentaenoic (DPA), and docosahexaenoic (DHA) acids.

The efficacy of the LC n-3 PUFAs in reducing risk for cardiovascular disease (CVD) remains controversial as findings from different randomized controlled trials (RCTs) have been conflicting. Nevertheless, a 2019 meta-analysis of RCTs reported significant reductions in risk for myocardial infarction, coronary heart disease (CHD) events and mortality, and CVD mortality in patients randomized to supplemental LC n-3 PUFAs[3]. Another meta-analysis of observational studies found that higher levels of circulating LC n-3 PUFA levels were significantly associated with a lower risk for CHD death[4]. However, no meta-analysis has yet examined the relationship between LC n-3 PUFAs blood levels and risk for all-cause mortality. Indeed, the only meta-analyses to report a beneficial association with all-cause mortality were based on the self-reported intake of fish[5,6]. Fish contain many nutrients besides just LC n-3 PUFAs, self-reported food intake is memory dependent, food databases can be out of date, and fish meals often replace less healthful choices. As a result, studies that link LC n-3 PUFAs and health outcomes based on self-reported fish intake have potential limitations. A more reliable and objective measure of LC n-3 PUFA consumption is their level in the blood[7] which is primarily determined by the consumption of preformed LC n-3 PUFAs (although synthesis from dietary ALA can make a small contribution[8]). Hence a clearer picture of the biological relationship between LC n-3 PUFAs and disease outcomes may be obtained from biomarker-based investigations.

Some studies have reported inverse relations between n-3 PUFA biomarkers and total mortality[9–11], while others have not[12,13]. In the Cardiovascular Health Study, higher LC n-3 PUFA levels also were associated with overall "healthier aging" (i.e., surviving past age 65 free of chronic diseases and maintaining good functional status)[14]. However, reports from studies of individual cohorts can be limited by insufficient power and inconsistent adjustment for potential confounding factors. In addition, publication bias can distort summary conclusions. To address these challenges, the present study pooled de novo individual-level analyses across 17 prospective cohort studies in the Fatty Acid and Outcome Research Consortium (FORCE)[15] to explore the associations of circulating levels of n-3 PUFAs (both plant- and seafood-derived) and all-cause mortality. Secondarily, we examined the associations with mortality from CVD, cancer, and all other causes.

Here, we show significant inverse associations for all mortality endpoints with the LC n-3 PUFA levels. Hence, chronically higher tissue levels of these FAs operating through a variety of potential mechanisms may slow the aging process.

## Results

**Population.** The pooled analyses included circulating n-3 PUFA measurements on 42,466 individuals, 15,720 (37%) of whom died during follow-up (Table 1). At baseline, the average age was 65 years (range of mean ages across cohorts was 50–81 years), 55% were women (range of 0–100% across cohorts) and the median follow-up time was 16 years (range of 5–32 years across cohorts). Whites constituted 87% of the sample. Circulating levels of the n-3 PUFAs (and of the n-6 PUFAs linoleic and arachidonic acids, which were included as covariates) are shown in Supplementary Fig. 1 and in Supplementary Table 2. Supplementary Table 3 shows the number of cause-specific deaths from participating cohorts. Overall, approximately 30% of the deaths were attributed to CVD, 30% to cancer, and the remaining 39% to all other causes.

**Total mortality.** Comparing the medians of the first and fifth quintiles (i.e., approximately the 90th and the 10th percentiles), higher EPA, DPA, DHA, and EPA + DHA levels were associated with between 9% and 13% lower risk of all-cause mortality (Table 2). (The fatty acid levels associated with these percentiles

**Table 1 Baseline characteristics[a] of 17 prospective cohort studies included in the meta-analysis: Fatty Acids and Outcomes Research Consortium.**

| Study | Country | Baseline year(s) | Follow-up years, median | N adults (N deaths) | Age, mean | Sex, % women | BMI, mean kg/m² | Lipid fraction |
|---|---|---|---|---|---|---|---|---|
| 60YO | Sweden | 1997–1999 | 19.5 | 3659 (756) | 60.0 | 52.0 | 26.7 | Plasma CE |
| AGES-R | Iceland | 2002–2006 | 9.4 | 1697 (962) | 76.9 | 55.2 | 27.2 | Plasma PL |
| CCCC | Taiwan | 1990–1991 | 18.9 | 1834 (993) | 60.6 | 44.0 | 23.3 | Plasma |
| CHS | United States | 1992–1993 | 13.3 | 2256 (1872) | 74.8 | 38.8 | 26.6 | Plasma PL |
| CSHA | Canada | 1991–1992 | 5.1 | 424 (19) | 80.9 | 61.0 | 25.8 | RBC PL |
| EPIC-Norfolk | United Kingdom | 1993–1997 | 17.4 | 6613 (3347) | 62.9 | 50.3 | 26.6 | Plasma PL |
| FHS | United States | 2008 | 7.3 | 2123 (292) | 65.4 | 56.6 | 28.3 | RBC PL |
| Hisayama | Japan | 2002 | 10.2 | 3293 (469) | 61.5 | 57.2 | 23.0 | Plasma |
| HPFS | United States | 1994 | 20.5 | 1477 (878) | 64.6 | 0.0 | 25.9 | RBC PL |
| KIHD | Finland | 1998–2001 | 17.9 | 1125 (310) | 61.8 | 48.3 | 27.4 | Plasma |
| MCCS | Australia | 1990–1994 | 23.2 | 3796 (902) | 54.5 | 54.8 | 26.9 | Plasma PL |
| MESA | United States | 2000–2002 | 14.0 | 1844 (111) | 69.8 | 5 | 28.4 | Plasma PL |
| MetSIM | Finland | 2006–2010 | 9.6 | 1354 (58) | 55.0 | 0.0 | 26.5 | Plasma PL |
| NHS | United States | 1989–1990 | 24.1 | 1487 (853) | 60.4 | 100 | 25.5 | RBC PL |
| 3C | France | 1999–2001 | 15.0 | 1421 (787) | 74.6 | 63.1 | 26.3 | Plasma |
| ULSAM | Sweden | 1970–1973 | 32.1 | 1878 (1771) | 49.7 | 0.0 | 25.0 | Plasma CE |
| WHIMS | United States | 1996 | 13.0 | 6185 (1340) | 70.1 | 100 | 28.4 | RBC PL |

[a]Baseline characteristics at the time of fatty acid biomarker measurement.

Abbreviations of cohorts: *60YO*, Stockholm cohort of 60-year olds, *AGES-R* Age, Genes, Environment Susceptibility Study (Reykjavik), *CCCC* Chin-Shan Community Cardiovascular Cohort Study, *CHS* Cardiovascular Health Study, *CSHA* Canadian Study of Health and Aging, *EPIC-Norfolk* European Prospective Investigation into Cancer, Norfolk UK, *FHS* Framingham Heart Study, *HPFS* Health Professionals Follow-up Study, *KIHD* Kuopio Ischemic Heart Disease Risk Factor Study, *MCCS* Melbourne Collaborative Cohort Study, *MESA* Multi-Ethnic Study of Atherosclerosis, *MetSIM* Metabolic Syndrome in Men Study, *NHS* Nurses' Health Study, *3C* Three-City Study, *ULSAM* Uppsala Longitudinal Study of Adult Men, *WHIMS* Women's Health Initiative Memory Study. CE cholesteryl esters, PL phospholipids, RBC red blood cells.

**Table 2 Associations of circulating n-3 PUFA biomarkers with risk of total and cause-specific mortality in 17 cohorts: Fatty Acids and Outcomes Research Consortium.**

| Fatty acid | All-cause mortality HR (95% CI) (17 cohorts; 15,720 deaths) | CVD mortality HR (95% CI) (15 cohorts; 4571 deaths) | Cancer mortality HR (95% CI) (15 cohorts; 4284 deaths) | Other mortality HR (95% CI) (14 cohorts; 6022 deaths) |
|---|---|---|---|---|
| ALA | 0.99 (0.96–1.02) | 1.01 (0.95–1.07) | 1.02 (0.96–1.08) | 0.99 (0.95–1.04) |
| EPA | 0.91 (0.88–0.94) | 0.88 (0.83–0.94) | 0.91 (0.85–0.96) | 0.92 (0.87–0.97) |
| DPA | 0.87 (0.84–0.91) | 0.91 (0.84–0.99) | 0.87 (0.81–0.95) | 0.88 (0.82–0.94) |
| DHA | 0.89 (0.85–0.92) | 0.86 (0.80–0.92) | 0.93 (0.86–1.00) | 0.90 (0.84–0.95) |
| EPA + DHA | 0.87 (0.83–0.90) | 0.85 (0.79–0.91) | 0.89 (0.83–0.96) | 0.88 (0.82–0.93) |

Hazard ratios (HRs) and 95% CIs expressed per cohort-specific inter-quintiles range comparing the midpoint of the top and bottom quintiles (see Supplementary Table 4 for cohort-specific n-3 PUFA values). All HRs are adjusted for age, sex, race, field center, body-mass index, education, occupation, marital status, smoking, physical activity, alcohol intake, prevalent diabetes, hypertension, and dyslipidemia, self-reported general health, and the sum of circulating n-6 PUFA (linoleic plus arachidonic acids). See Supplementary Table 4 for the 10th and 90th percentile values from each cohort for each PUFA of interest and the average PUFA values per lipid pool. *Abbreviations: ALA* alpha-linolenic acid, *CI* confidence interval, *CVD* cardiovascular disease, *DHA* docosahexaenoic acid, *DPA* docosapentaenoic acid, *EPA* eicosapentaenoic acid, *HR* hazard ratio.

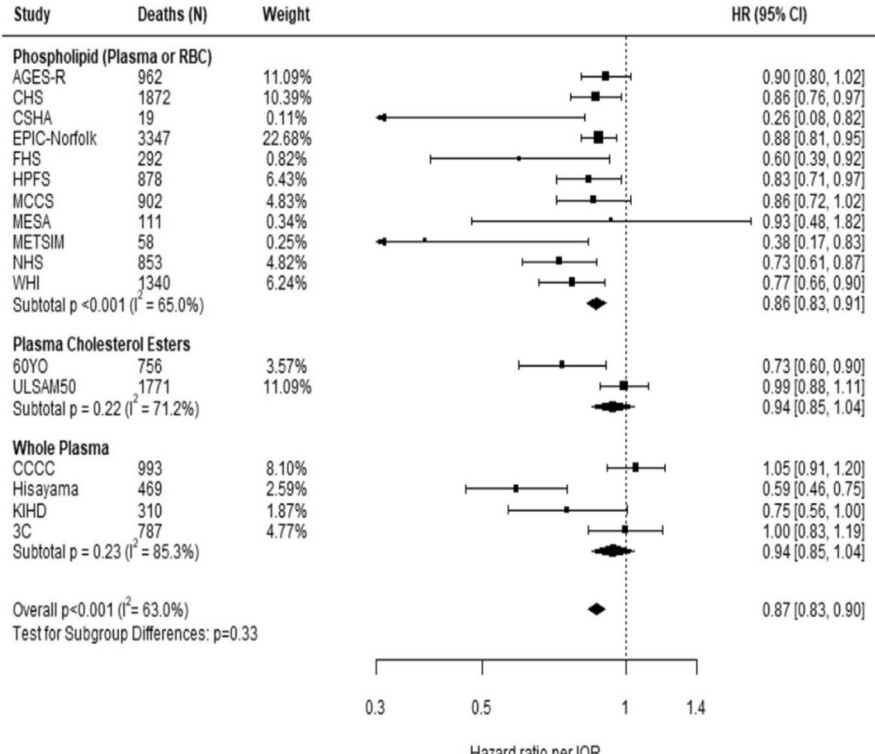

**Fig. 1 Adjusted hazard ratios (HR, 95% CI) for total mortality for circulating eicosapentaenoic (EPA) plus docosahexaenoic acid (DHA) in the 17 contributing studies of the Fatty Acids and Outcomes Research Consortium.** Study-specific estimates for HRs (dark squares) are shown per interquartile range (comparing the midpoint of the top to the bottom quintiles) their sizes indicate study weights (column 3). The horizontal line through each HR is 95% CI. Compartments included erythrocyte phospholipids, plasma phospholipids, cholesteryl esters, and total plasma. All HRs are adjusted for age, sex, race, field center, body-mass index, education, occupation, marital status, smoking, physical activity, alcohol intake, prevalent diabetes, hypertension, and dyslipidemia, self-reported general health, and the sum of circulating n-6 PUFA (linoleic plus arachidonic acids). See Table 1 footnote for abbreviations of cohorts.

for each cohort and sample type are shown in Supplementary Table 4). The HR for total mortality for EPA + DHA was 0.87 (95% CI: 0.83–0.90) (Fig. 1). In contrast, ALA was not significantly associated with all-cause mortality [HR 0.99 (0.96–1.02)]. In an across quintiles analysis, significant trends were observed for EPA, DPA, DHA, and EPA + DHA (all < 0.01); and comparing the top to the bottom quintile, each was associated with 15–18% lower risk of death (Table 3). There was little evidence for nonlinearity in these inverse associations for all each LC n-3 PUFAs except for EPA (p = 0.002 for the nonlinearity;

Fig. 2). The relationship of EPA with mortality was most pronounced at lower levels and then appeared to plateau at higher levels. ALA was generally unassociated with total mortality, except for a borderline association in the top quintile [HR 0.94 (0.89–0.99); *P*-trend = 0.13], and there was no evidence for nonlinearity (Supplementary Fig. 2).

**Cause-specific mortality.** Comparing the 90th to the 10th percentile, each of the LCn-3 PUFAs was significantly associated

**Table 3 Meta-analysis of circulating n-3 PUFA biomarkers with mortality types by cohort-specific quintiles (hazard ratios and 95% CIsᵃ): Fatty Acids and Outcomes Research Consortium.**

| Fatty acid | Quintiles | All-cause mortality (17 cohorts) | CVD mortality (15 cohorts) | Cancer mortality (15 cohorts) | Other mortality (14 cohorts) |
|---|---|---|---|---|---|
| ALA | Q1 | 1 | 1 | 1 | 1 |
| | Q2 | 0.95 (0.87-1.04) | 0.95 (0.87-1.04) | 0.98 (0.89-1.08) | 0.94 (0.87-1.01) |
| | Q3 | 0.94 (0.89-0.99) | 1.00 (0.91-1.10) | 0.96 (0.87-1.05) | 0.93 (0.86-1.00) |
| | Q4 | 0.95 (0.90-1.01) | 0.99 (0.91-1.09) | 0.99 (0.90-1.09) | 0.95 (0.88-1.03) |
| | Q5 | 0.94 (0.89-0.99) | 0.98 (0.89-1.08) | 0.88 (0.80-0.98) | 0.93 (0.86-1.01) |
| | *P* for Trendᵇ | 0.13 | 0.96 | 0.14 | 0.32 |
| EPA | Q1 | 1 | 1 | 1 | 1 |
| | Q2 | 0.92 (0.87-0.97) | 0.98 (0.90-1.07) | 0.90 (0.82-0.99) | 0.91 (0.84-0.98) |
| | Q3 | 0.88 (0.83-0.92) | 0.98 (0.90-1.07) | 0.86 (0.78-0.95) | 0.86 (0.79-0.93) |
| | Q4 | 0.85 (0.81-0.90) | 0.89 (0.81-0.98) | 0.87 (0.78-0.96) | 0.83 (0.77-0.90) |
| | Q5 | 0.82 (0.78-0.87) | 0.85 (0.77-0.94) | 0.82 (0.74-0.91) | 0.78 (0.72-0.85) |
| | *P* for Trend | <0.0001 | 0.006 | 0.008 | <0.0001 |
| DPA | Q1 | 1 | 1 | 1 | 1 |
| | Q2 | 0.95 (0.90-1.01) | 0.96 (0.87-1.07) | 0.96 (0.86-1.07) | 0.94 (0.86-1.02) |
| | Q3 | 0.92 (0.87-0.98) | 0.99 (0.89-1.09) | 0.98 (0.88-1.10) | 0.91 (0.84-0.99) |
| | Q4 | 0.90 (0.85-0.96) | 0.98 (0.88-1.09) | 0.92 (0.82-1.03) | 0.88 (0.80-0.96) |
| | Q5 | 0.84 (0.79-0.90) | 0.87 (0.78-0.98) | 0.79 (0.70-0.90) | 0.85 (0.78-0.93) |
| | *P* for Trend | 0.0001 | 0.16 | 0.008 | 0.007 |
| DHA | Q1 | 1 | 1 | 1 | 1 |
| | Q2 | 0.95 (0.90-1.00) | 0.96 (0.88-1.05) | 0.91 (0.83-1.00) | 0.96 (0.89-1.04) |
| | Q3 | 0.92 (0.88-0.97) | 0.87 (0.80-0.95) | 0.88 (0.80-0.97) | 0.97 (0.89-1.05) |
| | Q4 | 0.97 (0.94-1.01) | 0.92 (0.84-1.01) | 0.91 (0.83-1.00) | 0.90 (0.83-0.97) |
| | Q5 | 0.85 (0.81-0.90) | 0.79 (0.72-0.88) | 0.86 (0.78-0.95) | 0.87 (0.80-0.94) |
| | *P* for Trend | 0.01 | 0.002 | 0.06 | 0.008 |
| EPA + DHA | Q1 | 1 | 1 | 1 | 1 |
| | Q2 | 0.94 (0.89,0.99) | 0.96 (0.88,1.04) | 0.93 (0.85-1.03) | 0.93 (0.86-1.00) |
| | Q3 | 0.92 (0.88,0.97) | 0.91 (0.83,1.00) | 0.90 (0.82-0.99) | 0.94 (0.87-1.02) |
| | Q4 | 0.89 (0.84,0.93) | 0.86 (0.79,0.95) | 0.92 (0.83-1.02) | 0.89 (0.82-0.96) |
| | Q5 | 0.84 (0.79,0.89) | 0.80 (0.73,0.88) | 0.87 (0.78-0.96) | 0.82 (0.75-0.89) |
| | *P* for Trend | <0.0001 | 0.0004 | 0.06 | 0.0008 |

ᵃExpressed per cohort-specific quintiles (see Supplementary Table 4 for cohort-specific n-3 PUFA values). All hazard ratios are adjusted for age, sex, race, field center, body-mass index, education, occupation, marital status, smoking, physical activity, alcohol intake, prevalent diabetes, hypertension, and dyslipidemia, self-reported general health, and the sum of circulating n-6 PUFA (linoleic plus arachidonic acids).
ᵇP-for trend is computed by using a fixed-effects, inverse weighted meta-regression analysis, i.e., the hazard estimates were regressed against study quintiles, which we assigned a value of 1, 2, 3, 4, or 5.
*Abbreviations*: ALA alpha-linolenic acid, CI confidence interval, CVD cardiovascular disease, DHA docosahexaenoic acid, DPA docosapentaenoic acid, EPA eicosapentaenoic acid.

with a lower risk for death from CVD, cancer, and all other causes combined [except for DHA and cancer mortality, HR 0.93 (0.86–1.00)] (Table 2). ALA was not significantly associated with any cause-specific mortality. Evaluating the trend across quintiles, EPA, DHA, and EPA + DHA were inversely associated with CVD death, EPA and DPA were inversely associated with cancer death, and each of the LC n-3 PUFAs was inversely associated with other death. Comparing the top to the bottom quintile, EPA, DPA, DHA, and EPA + DHA were each significantly, inversely associated with CVD, cancer, and other mortality (Table 3).

**Heterogeneity and sensitivity analyses.** Inter-cohort heterogeneity was at least moderate ($I^2 > 50\%$) in the pooled analyses of all-cause mortality for all n-3 PUFAs except ALA ($I^2 = 26\%$) and EPA ($I^2 = 41\%$), while heterogeneity for cause-specific mortality ranged from little to moderate (0–56%) (Supplementary Table 5). There was little evidence of differential associations with mortality by PUFA lipid compartment after accounting for multiple testing (5 PUFAs × 4 outcomes; Bonferroni correction 0.05/20 = 0.0025, Supplementary Table 6). Likewise, associations of n-3 PUFAs with total mortality were similar across strata based on age, sex, race, and fish oil use (Supplementary Table 7), with no significant differences after accounting for multiple testing (5 PUFAs × 4 strata results; Bonferroni correction 0.05/20 = 0.0025). Overall findings did not change with the removal of

participants taking fish oil (Supplementary Table 7) or in the drop-one-cohort analyses.

## Discussion

In this meta-analysis utilizing a harmonized analytical strategy with individual-level data from 17 cohorts, we examined the associations between circulating levels of the n-3 PUFAs and mortality. We found that, after controlling for other major risk factors, LC n-3 PUFAs (but not ALA) were associated with about a 15–18% lower risk of total mortality comparing the top to the bottom quintiles. These relationships were generally linear for DPA, DHA, and EPA + DHA, but not for EPA. For this PUFA there was a steeper risk reduction across the lower blood levels but little additional difference in risk at higher blood levels. Inverse correlations were also generally observed between LC n-3 PUFA levels and CVD, cancer, and other causes of death.

This pooled analysis including over 40,000 participants and over 15,000 deaths greatly expands upon the findings of prior individual cohort studies that examined associations of circulating levels of n-3 PUFAs and all-cause mortality[9–13,16–24]. Relatively few studies have evaluated self-reported dietary fish (or estimated n-3 PUFA) intake in relation to total mortality, but those that have typically support our observations here[5,22,25,26]. Interestingly, reported use of fish oil supplements was linked to a

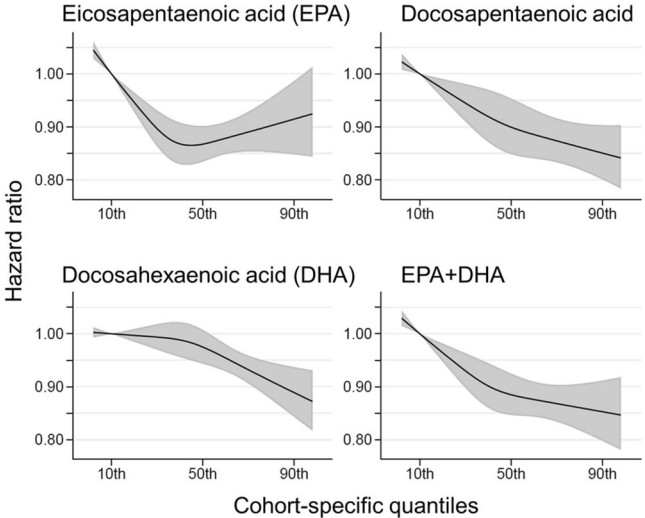

**Fig. 2 Associations of circulating long-chain n-3 PUFA levels with all-cause mortality: nonlinear dose-response meta-analysis in the Fatty Acids and Outcomes Research Consortium. Hazard ratios and cohort-specific quantiles are presented in the vertical and horizontal axis, respectively.** The best estimates and their confidence intervals are presented as black lines and gray-shaded areas, respectively. The 10th percentile was selected as a reference level and the x-axis depicts 5th to 95th percentiles. Potential nonlinearity was identified for EPA ($p = 0.0004$) but not for the others ($p > 0.05$). All HRs are adjusted for age, sex, race, field center, body-mass index, education, occupation, marital status, smoking, physical activity, alcohol intake, prevalent diabetes, hypertension, and dyslipidemia, self-reported general health, and the sum of circulating n-6 PUFA (linoleic plus arachidonic acids).

lower risk for death from any cause in a study from the UK including over 427,000 individuals[27].

Associations with total and cause-specific mortality were not significant for the plant-derived n-3 PUFA ALA. Prior biomarker-based meta-analyses reported inverse associations of ALA with CHD death, but relationships with total or CVD mortality were not examined[4,28]. Whether our finding of no association ALA on CVD mortality was because ALA has no role to play in fatal strokes (included in the CVD mortality metric) or because of differences in the cohorts included in these prior meta-analyses vs. the present one is not clear. Circulating ALA levels are less dependable markers of intake compared with the LC n-3 PUFAs because this fatty acid is rapidly β-oxidized and, to a small extent, converted into the LC n-3 PUFAs[8]. Nevertheless, the borderline and inconsistent relations of ALA on mortality risk deserve further study.

Higher circulating levels of LC n-3 PUFAs may beneficially affect diverse cellular systems that together could contribute to a reduced risk for death. The mechanisms behind the ostensibly beneficial effect of LC n-3 PUFAs on human biology are multiple and have been summarized in several recent reviews papers[29–32]. Among them are hypotriglyceridemic, antihypertensive, and antiplatelet effects; as well as positive effects on adipocyte biology, endothelial function, and autonomic balance. All of these appear to be mediated by effects on membrane physiochemistry, gene expression, and the production of a myriad of bioactive oxylipins. Persistently lower levels of inflammatory biomarkers also characterize those with higher circulating LC n-3 PUFA levels[33]. These fatty acids have been reported to inhibit the mammalian (or mechanistic) target of rapamycin (mTOR) in animal studies showing benefits in cancer[34], metabolic syndrome[35], spinal cord injury[36], and depression[37]. mTOR inhibition extends lifespan in

many species[38] and acts as an energy sensor to coordinate gene expression, ribosome biogenesis, and mitochondrial metabolism[39]. In the Heart and Soul Study, where whole blood EPA + DHA levels were inversely associated with all-cause mortality[24], higher levels were also linked with a slower rate of telomere shortening over a 5-year period[40]. As higher rates of telomere attrition have been associated with shorter overall lifespan[41,42], this finding may be secondary to the more distal biochemical mechanisms noted above. Regardless of their specific actions, higher cellular levels of the LC n-3 PUFAs appear to slow the aging process.

Our findings of lower risk of CVD death with high vs. low blood levels of EPA + DHA are generally consistent with meta-analyses of self-reported fish intake[25] and of biomarker levels[4], as well as randomized controlled clinical trials of n-3 PUFA supplementation[3,43] (although the most recent trial[44] has not yet been included in meta-analyses). Compared with CVD, evidence for a link between n-3 PUFAs and cancer mortality risk is sparse, with no significant relationship for self-reported estimates of fish or n-3 PUFA consumption[25,45]. Meta-analyses of RCTs with n-3 PUFA supplements also have not observed effects on cancer, although short-term durations of such trials (generally up to 5 years) would likely preclude any ability to detect an effect on cancer[46,47]. The difference between these findings and what we observed may arise from the use of biomarker levels instead of self-reported fish intake. Biomarkers are potentially truer reflections of long-term exposure, making it easier to detect subtle relationships. In addition, circulating LC n-3 PUFA levels reflect endogenous metabolism, especially for DPA which is not correlated with estimated dietary DPA intake[48] but may have important biologic effects[49]. Finally, since neurodegenerative diseases are a major non-CVD, non-cancer cause of death, a report that higher fish intake was associated with reduced mortality from this cause[6] is consistent with our observations here.

Although circulating marine n-3 PUFA levels have not been measured in all of the major intervention trials, the doses of EPA + DHA used in most trials (<1 g/day) may not have resulted in marked differences in levels between treated and control patients[50]. For example, in the Vitamin D and Omega-3 Trial (VITAL) trial, treatment with 840 mg of EPA + DHA per day increased plasma phospholipid EPA + DHA levels from 2.7 to 4.1%, a 55% increase. This relatively small difference in LC n-3 PUFA levels between the placebo and active treatment groups could be one of the potential reasons for the failure of some RCTs to detect an effect of n-3 PUFAs on CV outcomes[50,51]. Future RCTs may be more effective if they focus on people with low baseline levels of LC n-3 PUFAs[52] and provide doses of EPA and DHA that produce higher blood levels. An intake of about 250 mg of EPA + DHA per day as recommended in the Dietary Guidelines for Americans[53] may raise circulating levels into the ranges observed here for some but not all adults[7].

Although a significant effect on the primary (composite) endpoint in the VITAL trial[47] was not achieved, our findings comport well with some of its secondary findings. In this study, the provision of 840 mg of EPA + DHA/day significantly reduced risk for major CV events and myocardial infarction in those participants with lower (vs. higher) intakes of fish (blood levels in these groups were not reported). There was a significant interaction of fish intake on total mortality as well; the HR (95% CI) in the low intake group was 0.87 (0.73–1.04) and in the high intake group, 1.19 (0.99–1.44, $p$ for interaction 0.017). This secondary observation in VITAL implies that individuals with lower baseline LC n-3 PUFA levels are more likely to benefit from increased levels than those with higher baseline levels. Two recent RCTs examining the effects of high dose (~3–4 g/day) of LC n-3 PUFAs were performed in overweight patients with high blood triglyceride levels and at high risk for CVD events, all on background

statin therapy. After 5 years of treatment, Bhatt et al.[54] reported beneficial effects of EPA ethyl esters on CV events, whereas Nicholls et al.[44] found no effect on the primary outcome using an EPA + DHA product in which the fatty acids were non-esterified. Another 2-year trial in elderly post-MI patients from Norway given 1.8 g of EPA + DHA found no benefit on CV outcomes[55]. None of these trials is directly relevant to our findings here owing to the nature of the high-risk patient populations, the number of concurrent background medications, the short duration of treatment, and the initiation of treatment late in life.

Strengths of the current analysis include the use of objective n-3 PUFA biomarkers (instead of estimated intakes from dietary questionnaires) which increases the accuracy of exposure assessment and allows for separate analysis of different individual n-3 PUFAs. The use of prespecified, harmonized, de novo individual-level analyses across multiple cohorts substantially increase generalizability, reduces confounding through consistent adjustment for covariates, and limits the potential for publication bias. The pooling of 17 studies including over 15,000 deaths also increased the statistical power to evaluate mortality subtypes as well as potential heterogeneity across subgroups.

Potential limitations deserve attention. Because our outcome was not rare, the hazard ratios (HRs) reported here (instantaneous relative risk) may be modestly different than the cumulative relative risk. Most individuals were White, potentially lowering generalizability to other races/ethnicities, although our analysis still included nearly 6000 non-Whites in whom findings for EPA + DHA were generally similar to those for Whites (Supplementary Table 7). Despite extensive efforts to harmonize study-specific methods, moderate heterogeneity remained between studies that may be due to unmeasured background population characteristics, differences in laboratory assessment of PUFAs and of outcomes, chance, or any combination of these. PUFAs and covariates were measured once at baseline, and changes over time could lead to misclassification, which could bias the results in uncertain directions. On the other hand, reasonable reproducibility has been reported for n-3 PUFA biomarker concentrations over time[56]. Because analytical methods, even within the same lipid fraction, were not standardized, and n-3 PUFA levels were measured in multiple fractions, we assessed cohort-specific n-3 PUFA percentiles rather than absolute percentages of total fatty acids in each fraction. Since FA levels were reported as a percent of total FAs in each lipid compartment, levels of one FA could affect levels of another. Indeed, in the plasma or RBC PL and CE pools, higher levels of the LC n-3 PUFAs (which were the focus of this study) are linked with lower levels of the n-6 PUFAs but not of saturated or mono-unsaturated FAs[57,58]. Since we adjusted for differences in linoleic and arachidonic levels in our analyses, this concern was accounted for. Each lipid pool used in this study reflects LC n-3 PUFA intake during relatively different and overlapping time periods generally from months to weeks following this hierarchy: RBC ≥ Plasma PL ≈ Plasma CE ≥ total plasma[59,60]. In addition, we cannot rule out the potential for residual confounding. That is, higher LC n-3 PUFA levels may simply be markers of a "healthy lifestyle," and the fatty acids themselves may not be playing any physiological role in postponing death but would be biomarkers of a suite of other healthy behaviors (dietary/exercise/non-smoking, etc.), or endogenous metabolic processes, that might, in a multiplicity of ways, manifest in greater longevity. Although we adjusted for many major risk factors (age, income, marital status, smoking, hyperlipidemia, hypertension, etc.), residual confounding by other factors is always possible. However, the magnitude of the observed effect of the meta-analysis of circulating LC n-3 PUFAs and total mortality reported herein is consistent with the known associations with CHD mortality and sudden cardiac death[61,62].

Finally, as the attribution of cause of death is never as unambiguous as death itself, some uncertainty must attend to the cause-specific analyses reported here. In summary, in a global pooled analysis of prospective studies, LC n-3 PUFA levels were inversely associated with risk for death from all causes and from CVD, cancer, and other causes.

## Methods

**Study design and population: FORCE Consortium**. The study was conducted within FORCE[15], a consortium of observational studies with fatty acid biomarker data and ascertained chronic disease events[4]. For the current project, 48 prospective studies in the consortium as of December 2018 were invited to participate. Of these, seven did not have relevant data (e.g., no mortality outcomes or no circulating PUFA levels at baseline), two included only participants with prevalent CVD, 13 indicated a lack of funding/analyst time and 9 did not respond after at least 5 separate invitations to participate over a 9-month period. The study sample comprised data from 17 studies across 10 countries with available data on circulating PUFA levels at baseline and mortality during follow-up. The details of each individual study are presented in Supplementary Table 1. All participating studies followed a prespecified standardized analysis protocol with harmonized inclusions and exclusions, exposures, outcomes, covariates, and analytical methods including assessment of missing covariate data and statistical models. In each study, new analyses of individual data were performed according to the protocol, and study-specific results were collected using a standardized electronic form. Information regarding registration for any of the cohorts included herein (that required it prior to study initiation) is shown in Supplementary Table 1.

Individual cohorts conducted their studies in accordance with the criteria set by the Declaration of Helsinki, and informed consent was obtained from all participants. The review boards or ethics committees from each cohort were as follows: 60YO (Ethical Committee at the Karolinska Institut); AGES-R (Icelandic Heart Association and the Intramural Research Program of the National Institute on Aging); CCCC (National Taiwan University Research Ethics Committee); CHS (Tufts University Research Ethics Committee); CSHA (Laval University and the Research Center of the Centre Hospitalier Affilie Universitaire); EPIC-Norfolk (Norfolk Research Ethics Committee); FHS (Boston University Institutional Review Board); Hisayama (Kyushu University Certified Institutional Review Board); HPFS (Human Subjects Review Committee of the Harvard School of Public Health); KIHD (Research Ethics Committee of the University of Kuopio); MCCS (Cancer Council Victoria Human Research Ethics Committee); MESA (University of Washington Human Subjects Division); MetSIM (Ethics Committee of the University of Eastern Finland and Kuopio University Hospital); NHS (Human Research Committee at the Brigham and Women's Hospital); 3C (Consultative Committee for the Protection of Persons participating in Biomedical Research at Kremlin-Bicêtre University Hospital); ULSAM (Swedish Ethical Review Authority); and WHIMS (Fred Hutchinson Cancer Research Center Institutional Review Board).

Study participants in the included cohorts (a) were >18 years old, (b) had no major medical diagnoses (prior myocardial infarction, prior stroke, severe active cancer, severe renal disease, severe liver, or lung disease), (c) were not taking supplemental fish oil, and (d) did not die within a year of baseline. The one exception to (c) was the inclusion of the Age, Genes, Environment Susceptibility Study (Reykjavik) (AGES-R) from Iceland[63] in which 68% of participants reported taking cod liver oil. This factor was adjusted for in the AGES-R analysis, and participants in AGES-R taking cod liver oil were also excluded in a sensitivity analysis.

**Fatty acid measurements**. Participating studies measured PUFAs in at least one blood compartment, including plasma phospholipids, cholesterol esters, erythrocytes, and whole plasma. All PUFA levels were reported as a percent of total fatty acids. Detailed information regarding PUFA measurement methods for each study is in Supplementary Table 1.

**Outcome assessment**. The primary endpoint of this study was total mortality (death from any cause). Additional endpoints of interest were deaths from CVD, cancer, and all other causes. Detailed information on the definitions of the outcomes used in each cohort is included in Supplementary Table 1.

**Covariates**. Prespecified covariates included age (continuous), sex (men/women), race (binary: White/non-White), field center (categories), body-mass index (continuous), education (less than high school graduate, high school graduate, at least some college or vocational school), occupation (if available), marital status (married, never married, widowed, divorced), smoking (current, former, never), physical activity (kcal/week, METS/week, or hours/day), alcohol intake (drinks or servings/day, g/day or ml/day), prevalent diabetes mellitus (treated or physician-diagnosed), prevalent hypertension (treated or physician-diagnosed), prevalent dyslipidemia (treated or physician-diagnosed), self-reported general health (if available) and circulating n-6 PUFA levels (i.e., the sum of linoleic and arachidonic acids). If individual cohorts could not categorize these covariates exactly according

to these definitions, then study-specific categories were used as surrogates. Missing variables were handled as detailed in the Online Supplementary Materials.

**Statistical analysis and pooling**. Study-specific analyses were harmonized across cohorts. They were carried out using Cox proportional hazards models using robust variance estimates to calculate the multivariable-adjusted HRs in each study, with follow-up from the date of biomarker measurement to date of death, loss to follow-up, or end of follow-up. Associations and relevant statistical interactions were also assessed in prespecified strata within each cohort by age (<60 vs. ≥60), sex, and race (White vs. non-White). To allow comparison and pooling of results from different biomarker compartments, n-3 PUFA levels were standardized to the study-specific inter-quintiles range defined as the range between the medians of the top and bottom quintile categories (i.e., about the 90th and 10th percentiles). In addition, each cohort computed HRs across study-specific quintiles, with the lowest quintile as the reference. Pooling by quintiles instead of absolute fatty acid values were necessary because values differ by lipid compartment. Nevertheless, such an approach was reasonable given the observed correlations among different lipid compartments. For example, the Pearson correlations between EPA + DHA levels (i.e., percent of total fatty acids) in RBC and CE, PL, and whole plasma are 0.83, 0.88, and 0.93, respectively (unpublished data from Harris lab based on 49 samples analyzed in all four compartments).

*Meta-analysis*. Cohort-specific HRs were pooled by inverse-variance weighted meta-analysis. Heterogeneity was assessed by the $I^2$ statistic and Q-test. Heterogeneity was further explored by meta-analyzing prespecified subgroups. Sensitivity analyses included (1) the removal of those subjects from AGES-R who reported fish oil use, and (2) re-analysis after the removal of each cohort one at a time. The potential for a nonlinear association of each n-3 PUFA with all-cause mortality was examined with a multivariable meta-analysis with a restricted cubic spline technique[64] as detailed in Online Supplementary Materials. Stata 15.1 (Stata Corp., College Station, TX) was used for spline fitting and testing. All the other meta-analyses were conducted using the *metafor* package[65] in R version 3[66]. A two-tailed *P* value of <0.05 was considered to be statistically significant unless otherwise specified, e.g., in the exploratory analyses by subgroups and lipid compartments.

**Reporting summary**. Further information on research design is available in the Nature Research Reporting Summary linked to this article.

## Data availability
Policies for data-sharing vary between the cohorts depending on their original human subjects' approvals and existing procedures. For approved data-sharing requests, types of data that may be shared can include demographics, exposures, covariates, and outcomes. Please contact each individual principal investigator for cohort-specific data requests (See Supplementary Table 1).

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

## Author contributions

Cohort-associated personnel (i.e., principal investigators, associates, and trainees). I.A.B., K.-L.C., L.D., G.E., N.G.F., J.M.G., G.G.G., W.S.H., C.H., A.H., F.H., R.J., M.L., D.L., K.L., H.L., R.L., J.L., R.M., J.M., T.N., B.M.P., W.P., J.Q., J.G.R., U.R., V.G., K.-T.K., B.P., A.S.V., A.H.S., D.S.S., L.M.S., L.S., Q.S., Y.S., M.Y.T., A.M., J.H.Y.W., N.J.W., A.C.W., A.A.K., R.M.L., J.K.V., and D.M. Biostatistical analysts: J.K.B., P.-H.C., Y.-Y.C., Y.H., F.I., L.K.K., F.L., M.L., M.M., R.A.M., C.S., M.K.S., P.S., and N.T. Primary writing team: W.S.H., N.L.T., D.M., R.M.L., D.S.S., L.D., F.Q., and F.I.

## Competing interests

The authors below declare the following competing interests outside of the submitted work. A.I.B., Involvement in a research project partly funded by Unilever. A.S.V., Grants and support to attend professional meetings from the California Walnut Commission. B. M.P., Data and Safety Monitoring Board of a clinical trial funded by Zoll LifeCor; Steering Committee of the Yale Open Data Access Project funded by Johnson & Johnson. D.M., Research grants to Institution: the National Institutes of Health, the Gates Foundation, and the Rockefeller Foundation; Personal Fees: the Global Organization for EPA and DHA Omega-3, Bunge, Indigo Agriculture, Motif FoodWorks, Amarin, Acasti Pharma, Cleveland Clinic Foundation, Danone, and America's Test Kitchen; Scientific Advisory Boards: Brightseed, Calibrate, DayTwo, Elysium Health, Filtricine, Foodome, Human Co., and Tiny Organics; and Chapter Royalties: UpToDate. J.G.R., Research grants to Institution: Acasti, Amarin, Amgen, Astra-Zeneca, Eli Lilly, Esperion, Medicines Company, Merck, Novartis, Novo-Nordisk, Regeneron, and Sanofi. Consultant: Getz Pharma, Medicines Company, and Sanofi. R.A.M., Research grants to Institution: I. L.S.I. North America; Personal Fees from PharmaVite. The author below declares the following competing interests related to the submitted work. W.S.H., Stock in Omega-Quant Analytics, LLC (a laboratory that offers blood fatty acid testing); Schiff Institute Science and Innovation Advisory Board. The remaining authors declare no competing interests.

## Additional information

[1]Department of Internal Medicine, Sanford School of Medicine, University of South Dakota, Sioux Falls, SD, USA. [2]The Fatty Acid Research Institute, Sioux Falls, SD, USA. [3]Department of Mathematics and Statistics, Dordt University, Sioux Center, IA, USA. [4]MRC Epidemiology Unit, University of Cambridge School of Clinical Medicine, Cambridge, UK. [5]Department of Nutrition, Harvard T.H. Chan School of Public Health and Harvard Medical School, Boston, MA, USA. [6]Department of Medicine, Brigham and Women's Hospital, and Harvard Medical School, Boston, MA, USA. [7]The George Institute for Global Health and the Faculty of Medicine, University New South Wales, Sydney, Australia. [8]Department of Epidemiology, Johns Hopkins Bloomberg School of Public Health, Baltimore, MD, USA. [9]Cancer Epidemiology Division, Cancer Council Victoria, Melbourne, VIC, Australia. [10]Centre D'excellence Sur le Vieillissement de Québec, CIUSSS-CN, Quebec, QC, Canada. [11]Department of Medicine, Taipei Veterans General Hospital; and Institute of Epidemiology and Preventive Medicine, National Taiwan University, Taipei, Taiwan. [12]Department of Medicine and Clinical Science, Graduate School of Medical Sciences, Kyushu University, Fukuoka, Japan. [13]Division of Human Nutrition and Health, Wageningen University, Wageningen, The Netherlands. [14]Institute of Environmental Medicine, Karolinska Institute, Stockholm, Sweden. [15]Institute of Public Health and Clinical Nutrition, University of Eastern Finland, Kuopio, Finland. [16]Cancer Control Research, BC Cancer Agency; and School of Population and Public Health, University of British Columbia, Vancouver, BC, Canada. [17]Bordeaux Population Health Research Centre, INSERM, University of Bordeaux, Bordeaux, France. [18]USDA/ARS Children's Nutrition Research Center, Baylor College of Medicine, Houston, TX, USA. [19]Freidman School of Nutrition Science and Policy, Tufts University, Boston, MA, USA. [20]Department of Health Sciences, Faculty of Science, Vrije Universiteit Amsterdam, Amsterdam and Amsterdam Public Health Research Institute, Amsterdam, The Netherlands. [21]Institute of Epidemiology and Preventive Medicine, College of Public Health, National Taiwan University, Taipei, Taiwan. [22]Department of Internal Medicine, National Taiwan University Hospital, Taipei, Taiwan. [23]Icelandic Heart Association, Kopavogur, Iceland. [24]Centre for Epidemiology and Biostatistics, The University of Melbourne, Melbourne, Australia. [25]School of Health Sciences, University of Iceland, Reykjavík, Iceland. [26]Department of Internal Medicine, Division of Endocrinology, The Ohio State University, Columbus, OH, USA. [27]Institute of Clinical Medicine, University of Eastern Finland, Kuopio, Finland. [28]Department of Primary Care and Public Health, Imperial College, London, UK. [29]VITAM Research Centers, Laval University, Quebec, QC, Canada. [30]School of Epidemiology and Public Health, University of Ottawa, Ottawa, Canada. [31]Department of Epidemiology and Public Health, Kyushu University, Fukuoka, Japan. [32]Cardiovascular Health Research Unit, Department of Medicine, University of Washington, Seattle, WA, USA. [33]Department of Public Health and Caring Sciences, Clinical Nutrition and Metabolism, Uppsala University, Uppsala, Sweden. [34]Department of Medicine, University of Iowa, Iowa City, IA, USA. [35]Department of Epidemiology, University of Iowa, Iowa City, IA, USA. [36]Department of Family Medicine and Public Health, University of California San Diego School of Medicine, La Jolla, CA, USA. [37]Hospital del Mar Medical Research Institute & Barcelona βeta Brain Research Center, Pasqual Maragall Foundation, Barcelona, Spain. [38]Department of Preventive Medicine, University of Tennessee Health Science Center, Memphis, TN, USA. [39]School of Public Health, Division of Epidemiology and Community Health, University of Minnesota, Minneapolis, MN, USA. [40]Department of Laboratory Medicine and Pathology, University of Minnesota, Minneapolis, MN, USA. [41]The New York Academy of Medicine, New York, NY, USA. *A list of authors and their affiliations appears at the end of the paper.
✉email: wsh@faresinst.com

## The Fatty Acids and Outcomes Research Consortium (FORCE)

William S. Harris[1,2✉], Nathan L. Tintle[2,3], Fumiaki Imamura[4], Frank Qian[5,6], Andres V. Ardisson Korat[6], Matti Marklund[7,8], Luc Djoussé[6], Julie K. Bassett[9], Pierre-Hugues Carmichael[10], Yun-Yu Chen[11], Yoichiro Hirakawa[12], Leanne K. Küpers[13], Federica Laguzzi[14], Maria Lankinen[15], Rachel A. Murphy[16], Cécilia Samieri[17], Mackenzie K. Senn[18], Peilin Shi[19], Jyrki K. Virtanen[15], Ingeborg A. Brouwer[20], Kuo-Liong Chien[21,22], Gudny Eiriksdottir[23], Nita G. Forouhi[4], Johanna M. Geleijnse[13], Graham G. Giles[24], Vilmundur Gudnason[23,25], Catherine Helmer[17], Allison Hodge[24], Rebecca Jackson[26], Kay-Tee Khaw[4], Markku Laakso[27], Heidi Lai[19,28], Danielle Laurin[10,29], Karin Leander[14], Joan Lindsay[30], Renata Micha[19], Jaako Mursu[15], Toshiharu Ninomiya[31], Wendy Post[9], Bruce M. Psaty[32], Ulf Risérus[33], Jennifer G. Robinson[34,35], Aladdin H. Shadyab[36], Linda Snetselaar[35], Aleix Sala-Vila[2,37], Yangbo Sun[35,38], Lyn M. Steffen[39], Michael Y. Tsai[40], Nicholas J. Wareham[4], Alexis C. Wood[18], Jason H. Y. Wu[7], Frank Hu[5,6], Qi Sun[5,6], David S. Siscovick[41], Rozenn N. Lemaitre[32] & Dariush Mozaffarian[19]

