## [Peer Review File · Nature Communications]

Reviewers' Comments:

Reviewer #1:

Remarks to the Author:

This paper reports a meta-analytic study using individual data from large and very well-known observational studies to analyze the relationship of n-3 fatty acid levels and total and cause-specific mortality. I believe this is a very important topic that needs the special attention. The paper is well written, however, I think there are some important methodological issues they will need to improve in order to have a high quality individual-level meta-analytic study that can be used to develop future research and interventions based on their results. This reviewer consider that the authors have conducted good and adequate statistical analysis by what I can guess but I should not be guessing but really knowing the details of the analysis and, unfortunately with the information provided that is not very clear. The authors have missed some of the standard methodological procedures that should be conducted and reported in meta-analysis, there are several aspects that are not clear enough or they are not present at all or as they are, they will not allow to replicate the study. I think this paper can be improved methodologically if some more details are provided at least in supplemental material.

Here below I have some specific comments.

- Line 37: The authors should also refer to PRISMA and Cochrane Collaboration guidelines for individual-data meta-analysis.
- Line 40. It is not clear if all the studies with the data available accepted to participate or some of them were excluded or not accepted.
- Line 79. It is not clear how were the models develop for each study, were all using the same covariates? What software was used for this analysis? How was missing data handled at the study-level? As it is reported, it would be impossible to replicate those analysis.
- Line 82. Why were the authors using stratification by age and race instead of using those variables as possible covariates?
- I would recommend the authors to include citations on their statistical explanation.
- Line 89, how was the variance estimate?
- Line 90 and 94, important citations are missed here.
- Line 94, even reading the supplemental material is not detailed enough how the multivariable meta-analysis with restricted cubic spline technique was conducted. What variables were included, for example? What software or package in R was used?

Reviewer #2:

Remarks to the Author:

This is an interesting study investigating the association between circulating levels of plant- and marine derived n-3 PUFAs and the rate of all-cause death and mortality from CVD, cancer and other causes. The authors used data from the recognized FORCE Consortium. The paper is of high quality and well-written. However, some issues and limitations should be addressed in more detail.

Abstract

I suggest to rephrase the last sentence changing the word elevated e.g. "These novel findings suggest that high (compared to low) circulating levels of n-3 PUFA may be associated with a lower risk of premature death."

Introduction

The authors mention that ALA can make a contribution to levels of LC n-3 PUFAs (line 22). Indeed, the relative conversion of ALA appears to be limited in humans with reported conversions of ALA to EPA of less than 10% and DHA of less than 1% (Baker et al. Progress in Lipid Research 2016). The intake of ALA is, however, markedly higher than of marine n-3 PUFAs in western countries. Could the absolute contribution of predominantly EPA from ALA therefore be of importance due to the often substantially higher intake of ALA than of marine n-3 PUFAs?

Methods

I absolutely agree that the use of biomarkers should be encouraged in studies of (marine) n-3 fatty acids. The authors combined plasma, CE+PL and RBC as biomarkers for n-3 intake. However, the content of n-3 PUFAs in RBCs may reflect the exposure to n-3 PUFAs for a longer period than the content in plasma CE and phospholipids. Further, plasma is less suitable than phospholipids for use as biomarker partly because of a high variability in day-to-day measurement of plasma fatty acid composition. The authors should acknowledge these issues in the discussion and could mention that the content of n-3 PUFAs in adipose tissue might be a better biomarker because it may reflect the long-term exposure to these fatty acids.

Validation of cases is important. While myocardial infarction cases are usually registered with a high positive predictive value, this is more difficult and uncertain for sudden cardiac death and stroke. This should be acknowledged and commented

A major advantage of this study was the standardized analysis with a harmonized pre-specified analytical approach. The authors included adjustment for hypertension and dyslipidemia, but acknowledge in the discussion that LC n-3 PUFAs may exert hypotriglyceridemic and anti-hypertensive effects among others (line 166-167). Therefore, these comorbidities could be seen as potential intermediates in the causal pathway between n-3 exposure and cardiovascular death. The authors should comment on this.

The authors reported their associations of interest in risk percentage e.g. "Comparing the top to the bottom quintile, EPA, DPA, DHA, and EPA+DHA were each associated with 14-17% lower risk of total mortality" (line 114-116). However, a large proportion of the study population(s) died during follow-up, which may challenge the interpretation of a hazard rate ratio as a relative risk. Please comment on this.

The authors investigated the non-linear association between n-3 PUFAs and their outcomes of interest, which is a major strength of this study. Please comment on why the categorical analyses were chosen as primary analyses over the continuous restricted cubic spline analyses. In Stata version 16, the knots using restricted cubic splines are located at the 10th, 50th and 90th percentile by default in accordance with the recommendations by Harrell (Regression Modeling Strategies: with Applications to Linear Models, Logistic and Ordinal Regression, and Survival Analysis, Second Edition, Springer 2015). Please comment on why the knots were placed at the 25th, 50th and 75th percentile as this may imply that a larger part of the curves are linear.

Results

These are clear and convincing. Thus, all point estimates in Fig 1 (apart from the CCCC study) goes in the same direction (a reduction in death) which I think the authors should mention. Death from other causes was the largest contributor to death (39%) and the authors might consider mentioning a few of the most important causes. The authors might consider including data for ALA in Fig 2 instead of a separate Fig S2. Also, I believe data on use of fish oil concentrates could be removed from Table S7 because this was generally an exclusion criteria and nothing is reported on doses or intake of seafood in these participants. Fig S1 is of interest, but difficult to read, I suggest colour codes.

Discussion

The authors have addressed limitations of their study in detail. A major issue is that biomarkers were only determined at baseline which is mentioned by the authors.

DPA is a somewhat controversial issue. While effects of EPA and DHA are established they are less documented for DPA (one reference is given). Could the authors speculate from what sources the measured DPA is derived (seafood? Meat? Other food? Metabolism of EPA and DHA?).

The authors are encouraged to avoid the wording "effect" (e.g. lines 157, 162, 181 and 184) when referring to pattern of associations as this phraseology may insinuate causality.

Despite the fact that the authors did not investigate fish consumption or fish oil supplements per se, I suggest that the authors discuss their results in the context of current guidelines.

Changes to the manuscript that were not requested

During the period of time between the original submission and this opportunity to revise, two new cohorts - the Nurses' Health Study and the Health Professional Follow-up study - provided data for this paper. These data are now included, and all tables and figures are updated to include these two new cohorts. Their inclusion only strengthened the original conclusions.

The title was changed from a "meta-analysis" to a "de novo pooled analysis." This was done to distinguish this study from the standard meta analysis that uses only published data. Most of the studies included in the present analysis have not been published.

Owing to the addition of two new cohorts additional authors have been added and the order of the authors has been slightly shifted.

We made an addition to the methods section to describe how many cohorts were approached for potential participation in this project and how many eventually did so, along with reasons why some of those not responding did not or were unable to participate.

We have included a mention of the correlations for omega-3 levels between lipid pools using unpublished data from my laboratory.

We added a short discussion of an additional mechanism of action by which omega-3 fatty acids may prolong life, i.e., by inhibiting mTor signaling.

We have also included an updated reference to the STRENGTH randomized trial that was reported out during the review process, and to a new meta analysis of randomized trials by Bernasconi et al.

Finally, the primary affiliation of the corresponding author has been changed from OmegaQuant Analytics to the Fatty Acid Research Institute.

REVIEWER COMMENTS

Reviewer #1 (Remarks to the Author):

This paper reports a meta-analytic study using individual data from large and very well-known observational studies to analyze the relationship of n-3 fatty acid levels and total and cause-specific mortality. I believe this is a very important topic that needs the special attention. The paper is well written, however, I think there are some important methodological issues they will need to improve in order to have a high quality individual-level meta-analytic study that can be used to develop future research and interventions based on their results. This reviewer consider that the authors have conducted good and adequate statistical analysis by what I can guess but I should not be guessing but really knowing the details of the analysis and, unfortunately with the information provided that is not very clear. The authors have missed some of the standard methodological procedures that should be conducted and reported in meta-analysis, there are several aspects that are not clear enough or they are not present at all or as they are, they will not allow to replicate the study. I think this paper can be improved methodologically if some more details are provided at least in supplemental material. Here below I have some specific comments.

- Line 37: The authors should also refer to PRISMA and Cochrane Collaboration guidelines for individual-data meta-analysis.

We have reviewed these guidelines and, in conjunction with our responses to specific reviewer comments (see subsequent responses below) have provided additional information online with PRISMA/Cochrane guidelines.

- Line 40. It is not clear if all the studies with the data available accepted to participate or some of them were excluded or not accepted.

We have added additional information on how cohorts were invited and responded to our invitation. We will note that the primary reason for lack of participation was a lack of time/funding. During the time between submission and this invitation to resubmit, two of the cohorts initially indicating that they could not participate (due to a lack of funds to support the analysts, i.e., the Nurse's Health Study and Health Professionals Follow-up Study) reached out to us indicating they now had time to participate. They have now been added to the analysis, and all data throughout the manuscript has been updated accordingly. The addition of these two additional cohorts had no substantive impact on any of the findings other than to slightly strengthen them. Due to the sheer volume of tiny changes in the values in the Tables, these changes are not marked in the manuscript.

- Line 79. It is not clear how were the models develop for each study, were all using the same covariates? What software was used for this analysis? How was missing data handled at the study-level? As it is reported, it would be impossible to replicate those analysis.

Covariates were all handled similarly by the individual cohorts as they analyzed the data for their specific study. We have added the covariates list to the Study-specific analyses section (prior it was only listed in a footnote in the results). For missing covariate data at the cohort level, a missing indicator category was used for categorical covariates. Missing continuous covariates were handled per the usual practice of each cohort and study investigators, e.g., imputation or exclusion. Missing data handling at the cohort-specific level is noted on page 18 of the supplemental materials (and referenced as such in the methods section of the main manuscript).

- Line 82. Why were the authors using stratification by age and race instead of using those variables as possible covariates?

All hazard ratios were adjusted for age, sex, race, field center, body-mass index, education, occupation, marital status, smoking, physical activity, alcohol intake, prevalent diabetes, hypertension, and dyslipidemia, self-reported general health, and the sum of circulating n-6 PUFA (linoleic plus arachidonic acids). We also then did stratified analyses by age, sex and race. We have updated the methods section text to clarify these points.

- I would recommend the authors to include citations on their statistical explanation.

Additional citations have been added to the main text as well as to the Supplemental Materials where additional details are provided.

- Line 89, how was the variance estimate?

Primary analyses were conducted using a fixed effects approach. We also conducted random effects models using a restricted maximum likelihood estimator and observed no substantial difference in findings (detailed results not shown). We have added these clarifications to the Online Supplemental Materials which have been expanded with regards to the meta-analysis strategy.

- Line 90 and 94, important citations are missed here.

Additional citations have been added to the primary methods section as well as to the Online Supplemental Materials.

- Line 94, even reading the supplemental material is not detailed enough to understand how the multivariable meta-analysis with restricted cubic spline technique was conducted. What variables were included, for example? What software or package in R was used?

We have expanded our explanation of the spline methodology in the supplemental materials. As noted in the methods section Stata was used for spline fitting and testing. The method is consistent with the one proposed by Orsini et al. (1) who provided the Stata code in Stata J (2). We have used them after minor modification for our results. We have included this point in our explanation, too. For a standard meta-analysis, the metafor package in R was used.

Reviewer #2 (Remarks to the Author):

This is an interesting study investigating the association between circulating levels of plant- and marine derived n-3 PUFAs and the rate of all-cause death and mortality from CVD, cancer and other causes. The authors used data from the recognized FORCE Consortium. The paper is of high quality and well-written. However, some issues and limitations should be addressed in more detail.

Abstract

I suggest rephrasing the last sentence, changing the word “elevated” e.g. “These novel findings suggest that high (compared to low) circulating levels of n-3 PUFA may be associated with a lower risk of premature death.”

We have changed it to say, “These novel findings suggest that higher circulating levels of n-3 PUFA may be associated with a lower risk of premature death.” The word “higher” implies comparison with “lower” and is consistent with “a lower risk...”

Introduction

The authors mention that ALA can make a contribution to levels of LC n-3 PUFAs (line 22). Indeed, the relative conversion of ALA appears to be limited in humans with reported conversions of ALA to EPA of less than 10% and DHA of less than 1% (Baker et al. Progress in Lipid Research 2016). The intake of ALA is, however, markedly higher than of marine n-3 PUFAs in western countries. Could the absolute contribution of predominantly EPA from ALA therefore be of importance due to the often substantially higher intake of ALA than of marine n-3 PUFAs?

It's a good question you've raised. In reviewing the Baker article it is clear ALA can be converted to EPA, but based on Figure 4 it seems highly unlikely that ALA in practice contributes much EPA at all. The average ALA intake in the US is about 1.6 g per day. Let's say someone is eating 2x this amount, or about 3 g per day. That person's EPA level might increase by 20% (per Fig 4). That would increase it from around 0.3% of plasma phospholipid fatty acids to 0.36%. This small difference would be about the limit of detection in the methods used to analyze FA composition. So, in the end, it's unlikely that ALA is contributing much to the EPA value here.

Methods

I absolutely agree that the use of biomarkers should be encouraged in studies of (marine) n-3 fatty acids. The authors combined plasma, CE+PL and RBC as biomarkers for n-3 intake. However, the content of n-3 PUFAs in RBCs may reflect the exposure to n-3 PUFAs for a longer period than the content in plasma CE and phospholipids. Further, plasma is less suitable than phospholipids for use as biomarker partly because of a high variability in day-to-day measurement of plasma fatty acid composition. The authors should acknowledge these issues in the discussion and could mention that the content of n-3 PUFAs in adipose tissue might be a better biomarker because it may reflect the long-term exposure to these fatty acids.

These are good points, and you're right that RBCs are a better long-term marker for omega-3 status than really any other lipid compartment except perhaps (as you note) adipose tissue. None of the studies available to us used adipose tissue and it's obviously not a clinically practical sample type to collect and analyze. Stepping back for a moment and taking a look at the bigger picture of what we are trying to accomplish may be helpful. When trying to compare populations for “circulating omega-3 status” where status has been measured in different lipid pools, we have assumed that even though

there may be more day to day noise in non-RBC omega-3 levels, when pooled across hundreds of patients, very clear patterns emerge. Data from our lab has shown that the correlations (for EPA+DHA) between RBC and CE, PL and whole plasma are 0.83, 0.93 and 0.87, respectively. So, although the values are different, the correlations are high. And so, when we rank people by quintile within each cohort, the quintile rankings can be used to pool all of data across the 17 cohorts. We have now noted these correlations in the Statistical Analysis section to highlight this point.

Validation of cases is important. While myocardial infarction cases are usually registered with a high positive predictive value, this is more difficult and uncertain for sudden cardiac death and stroke. This should be acknowledged and commented

We were not that detailed in this analysis. Obviously, death from any cause is completely unambiguous. Death from CVD, from cancer or from anything else is perhaps not quite as clean, but it does not require differentiating SCD from acute, massive strokes as the cause of a CVD death.

A major advantage of this study was the standardized analysis with a harmonized pre-specified analytical approach. The authors included adjustment for hypertension and dyslipidemia, but acknowledge in the discussion that LC n-3 PUFAs may exert hypotriglyceridemic and anti-hypertensive effects among others (line 166-167). Therefore, these comorbidities could be seen as potential intermediates in the causal pathway between n-3 exposure and cardiovascular death. The authors should comment on this.

Yes, it's true that LC n-3 PUFA – when given at 'pharmacologic' doses (3+ g per day) - do have clear TG-lowering and relatively clear BP-lowering effects. However, virtually none of the people in these studies were taking fish oil supplements, especially at doses this high. (A typical fish oil consumer will take 1 capsule containing 300 mg of EPA+DHA... nowhere near enough to affect TG or BP). One cohort was taking fish oil routinely, however, the AGES cohort from Iceland where cod liver oil is very commonly taken. Our sensitivity analysis found that removing AGES did not altering the overall outcome. This suggests that omega-3 intakes were not high enough to meaningfully alter TG or BP, and thus allowing dyslipidemia (which more commonly means high cholesterol than high TG) and hypertension to remain as covariates.

The authors reported their associations of interest in risk percentage e.g. "Comparing the top to the bottom quintile, EPA, DPA, DHA, and EPA+DHA were each associated with 14-17% lower risk of total mortality" (line 114-116). However, a large proportion of the study population(s) died during follow-up, which may challenge the interpretation of a hazard rate ratio as a relative risk. Please comment on this.

We have noted in the limitations section that caution should be taken when interpreting the hazard rate as a relative risk due to the high percent mortality (37%).

The authors investigated the non-linear association between n-3 PUFAs and their outcomes of interest, which is a major strength of this study. Please comment on why the categorical analyses were chosen as primary analyses over the continuous restricted cubic spline analyses. In Stata version 16, the knots using restricted cubic splines are located at the 10th, 50th and 90th percentile by default in accordance with the recommendations by Harrell (Regression Modeling Strategies: with Applications to Linear Models, Logistic and Ordinal Regression, and Survival Analysis, Second Edition, Springer 2015). Please comment on why the knots were placed at the 25th, 50th and 75th percentile as this may imply that a larger part of the curves are linear.

We appreciate the point and have now revised the figures by using the 10th, 50th, and 90th percentile values as knots. After this revision and new datasets from two cohorts (NHS and HPFS), the figures have changed visually to some extent, but we have seen no need to change our overall interpretation (non-linearity for EPA with $p=0.0004$, linearity for the others with $p>0.05$).

The rationale of our prior decision is here: In our harmonized meta-analysis, each participating cohort provided point estimates of quintile groups. Accordingly, our data points of the exposure (each fatty acid) were median values of all the quintile groups. By design, therefore, data points from minimal value to 10th percentiles would be a few, as well as data points from the 90th percentile to the maximal value. Nonetheless, given the large number of our participating cohorts, it appeared that we could model the spline curve by using the 10th, 50th, and 90th percentiles. Thus, we have revised the figures according to the recommendation.

Results

These are clear and convincing. Thus, all point estimates in Fig 1 (apart from the CCCC study) goes in the same direction (a reduction in death) which I think the authors should mention. Death from other causes was the largest contributor to death (39%) and the authors might consider mentioning a few of the most important causes.

We thought long and hard about the “other causes” while writing this paper. We sought input from the cohorts – few of which uniformly collected data on the same causes of death – and tried to piece together some hints about which causes were involved. We finally decided that this paper was long enough and complicated enough that to dig more deeply than the three “causes” we did report would have to wait for another paper, another day.

The authors might consider including data for ALA in Fig 2 instead of a separate Fig S2.

The journal has a restriction that there can be at most 4 panels within a figure, and so we decided to leave ALA in Figure S2.

Also, I believe data on use of fish oil concentrates could be removed from Table S7 because this was generally an exclusion criteria and nothing is reported on doses or intake of seafood in these participants.

Yes, taking fish oil was generally an exclusion criterion, but we stipulated in the study protocol that any cohort with a very high usage of fish oil would be included since that’s normative for that population. Nevertheless, we planned ahead to do a sensitivity analysis eliminating any such cohort to determine its effect on the overall result. This is what happened with AGES-R (Iceland). So we need to keep this in table S7. And yes, we were not particularly interested in collecting the dietary data (of highly questionable accuracy) on seafood intake since we were able to do our analysis directly with the biomarker itself - a much more satisfactory approach to answering our study question.

Fig S1 is of interest, but difficult to read, I suggest color codes.

Figure S1 now has a color coded legend.

Discussion

DPA is a somewhat controversial issue. While effects of EPA and DHA are established they are less documented for DPA (one reference is given). Could the authors speculate from what sources the measured DPA is derived (seafood? Meat? Other food? Metabolism of EPA and DHA?).

We have now cited a 2019 review paper by Richter et al indicating that its levels are *not* correlated with intake, and hence are metabolically determined.

The authors are encouraged to avoid the wording “effect” (e.g. lines 157, 162, 181 and 184) when referring to pattern of associations as this phraseology may insinuate causality.

Yes, this is usually something we try to be careful about. We have now cleansed the paper of this language except where it applies to intervention or experimental studies where ‘effects’ actually applies.

Despite the fact that the authors did not investigate fish consumption or fish oil supplements per se, I suggest that the authors discuss their results in the context of current guidelines.

The current guidelines in the US – Dietary Guidelines for Americans – recommends consuming oily fish so as to achieve a 250 mg/d intake of EPA+DHA. Although this would be a doubling of current intakes, it may not be sufficient to achieve the highest omega-3 levels where we observed the greatest risk reductions. This has now been noted in the Discussion.

Editorial Comments

We have now included a statement regarding data availability at the end of the manuscript:

Data Availability statement

The data that support the findings of this study are available from the individual cohorts that agreed to participate in this harmonized meta-analysis, but restrictions apply to the availability of these data, which were used under agreement for the current study, and so are not publicly available. Data are however available from the authors upon reasonable request and with permission of the individual cohorts (see Table S1).

Reviewers' Comments:

Reviewer #1:

Remarks to the Author:

I believe the authors have addressed all the methodological concerns and I would consider this review for publication.

Reviewer #3:

Remarks to the Author:

The authors have carefully addressed my questions raised and the inclusion of data from the Nurses' Health Study and the Health Professional Follow-up Study is a significant contribution to the manuscript. I have a few additional comments:

The authors have cited the STRENGTH trial in this revised manuscript as this was not included in the meta-analysis by Bernasconi et al. (reference 47). The authors may consider also to cite the recently published OMEMI trial (PMID: 33191772), which investigated supplementation with marine n-3 PUFAs on cardiovascular events in elderly.

The authors reported all PUFA levels as percent of total fatty acids, which is the most common approach in biomarker studies using fatty acids as exposure. However, the relative content of any individual fatty acid measured thus may depend on the content of other fatty acids. The authors should address this limitation in the discussion. Also, circulating levels of n-3 PUFAs may be considered shorter-term biomarkers reflecting exposure from days to (few) months. The authors should comment on how long their circulating biomarker concentrations may reflect n-3 exposure in the discussion.

The spline analyses conducted are very informative. However, in Figure 2 the X-axis does not seem to reflect the whole exposure width (0 to 100th percentile). The authors should add to the Figure legend in case the X-axis does not reflect the whole exposure width.

In the online supplement, the authors reported that fifty nine percent of the study participants in the Health Professional Follow-up Study and seventy one percent in the Nurses' Health Study provided fasting blood samples. However, less than 1500 participants from these very large cohorts were included in presented analyses. Please comment on this.

Reviewer #3:

Comment 1. The authors have cited the STRENGTH trial in this revised manuscript as this was not included in the meta-analysis by Bernasconi et al. (reference 47). The authors may consider also to cite the recently published OMEMI trial (PMID: 33191772), which investigated supplementation with marine n-3 PUFAs on cardiovascular events in elderly.

Now cited in Discussion. “Another 2-year trial in elderly post-MI patients from Norway given 1.8 g of EPA+DHA found no benefit on CV outcomes.(1)”

Comment 2. The authors reported all PUFA levels as percent of total fatty acids, which is the most common approach in biomarker studies using fatty acids as exposure. However, the relative content of any individual fatty acid measured thus may depend on the content of other fatty acids. The authors should address this limitation in the discussion.

The following has been added to the limitations section: “Since FA levels were reported as a percent of total FAs in each lipid compartment, levels of one FA could affect levels of another. Indeed, in the plasma or RBC PL and CE pools, higher levels of the LC n-3 PUFAs (which were the focus of this study) are linked with lower levels of the n-6 PUFAs but not of saturated or mono-unsaturated FAs.(2, 3) Since we adjusted for differences in linoleic and arachidonic levels in our analyses, this concern was accounted for.”

Comment 3. Also, circulating levels of n-3 PUFAs may be considered shorter-term biomarkers reflecting exposure from days to (few) months. The authors should comment on how long their circulating biomarker concentrations may reflect n-3 exposure in the discussion.

The following has been added to the Limitation section: “Each lipid pool used in this study reflects LC n-3 PUFA intake during relatively different and overlapping time periods generally from months to weeks following this hierarchy: RBC ≥ Plasma PL ≈ Plasma CE ≥ total plasma (4, 5)

Comment 4. The spline analyses conducted are very informative. However, in Figure 2 the X-axis does not seem to reflect the whole exposure width (0 to 100th percentile). The authors should add to the Figure legend in case the X-axis does not reflect the whole exposure width.

The legend has been modified to define the limits of the X-axis.

Comment 5. In the online supplement, the authors reported that 59 percent of the study participants in the Health Professional Follow-up Study and 71 percent in the Nurses' Health Study provided fasting blood samples. However, less than 1500 participants from these very large cohorts were included in presented analyses. Please comment on this.

Interesting point. To my understanding, that means that although there were lots of fasting plasma samples that could have been analyzed for fatty acid composition, at the time the analyses were undertaken, the investigators only had resources to pay for 1500 or that they only needed 1500 to examine whatever outcome was the focus of the ancillary project that they were funded to do. This does not change the fact that the 1500 we used in this study met all study criteria.

References

1. Kalstad AA, Myhre PL, Laake K, Tveit SH, Schmidt EB, Smith P, Nilsen DWT, Tveit A, Fagerland MW, Solheim S, et al. Effects of n-3 Fatty Acid Supplements in Elderly Patients after Myocardial Infarction: A Randomized Controlled Trial. *Circulation* 2020.
2. Flock MR, Skulas-Ray AC, Harris WS, Etherton TD, Fleming JA, Kris-Etherton PM. Determinants of Erythrocyte Omega-3 Fatty Acid Content in Response to Fish Oil Supplementation: A Dose-Response Randomized Controlled Trial. *Journal of the American Heart Association* 2013;2(6):e000513.
3. Young AJ, Marriott BP, Champagne CM, Hawes MR, Montain SJ, Johannsen NM, Berry K, Hibbeln JR. Blood fatty acid changes in healthy young Americans in response to a 10-week diet that increased n-3 and reduced n-6 fatty acid consumption: a randomised controlled trial. *The British journal of nutrition* 2017:1-13.
4. Arab L. Biomarkers of fat and fatty acid intake. *J Nutr* 2003;133 Suppl 3:925S-32S.
5. Hodson L, Skeaff CM, Fielding BA. Fatty acid composition of adipose tissue and blood in humans and its use as a biomarker of dietary intake. *Prog Lipid Res* 2008;47:348-80.

Reviewers' Comments:

Reviewer #3:

Remarks to the Author:

The authors have carefully addressed all my questions. I have no further comments.